

# Estimating the effect of burrowing shrimp on deep-sea sediment community oxygen consumption

Daniel Leduc[1] and Conrad A. Pilditch[2]

[1] National Institute of Water and Atmospheric Research (NIWA), Wellington, New Zealand
[2] School of Science, University of Waikato, Hamilton, New Zealand

## ABSTRACT

Sediment community oxygen consumption (SCOC) is a proxy for organic matter processing and thus provides a useful proxy of benthic ecosystem function. Oxygen uptake in deep-sea sediments is mainly driven by bacteria, and the direct contribution of benthic macro- and mega-infauna respiration is thought to be relatively modest. However, the main contribution of infaunal organisms to benthic respiration, particularly large burrowing organisms, is likely to be indirect and mainly driven by processes such as feeding and bioturbation that stimulate bacterial metabolism and promote the chemical oxidation of reduced solutes. Here, we estimate the direct and indirect contributions of burrowing shrimp (*Eucalastacus* cf. *torbeni*) to sediment community oxygen consumption based on incubations of sediment cores from 490 m depth on the continental slope of New Zealand. Results indicate that the presence of one shrimp in the sediment is responsible for an oxygen uptake rate of about 40 $\mu$mol d$^{-1}$, only 1% of which is estimated to be due to shrimp respiration. We estimate that the presence of ten burrowing shrimp m$^{-2}$ of seabed would lead to an oxygen uptake comparable to current estimates of macro-infaunal community respiration on Chatham Rise based on allometric equations, and would increase total sediment community oxygen uptake by 14% compared to sediment without shrimp. Our findings suggest that oxygen consumption mediated by burrowing shrimp may be substantial in continental slope ecosystems.

## INTRODUCTION

Deep-sea soft sediment communities play an important role in global carbon cycling (*Jahnke & Jackson, 1992*, *Archer & Maier-Reimer, 1994*). The input of particulate organic carbon (POC) from surface waters is the main driver of benthic metabolism in deep-sea sediments (*Smith, 1987*; *Pfannkuche, 1993*), which in turn is influenced by surface (e.g., seasonal and inter-annual variability in climate; *Lampitt et al., 2001*; *Smith et al., 2006*), and water column processes (e.g., hydrodynamics, POC recycling and remineralisation by bacteria; *Lampitt & Antia, 1997*; *Turner, 2002*). Processing of organic material and overall metabolism in deep-sea sediments are assumed to be dominated by bacteria and small fauna

Corresponding author
Daniel Leduc,
daniel.leduc@gmail.com,
Daniel.Leduc@niwa.co.z

(e.g., *Schwinghamer et al., 1986*; *Pfannkuche, 1993*; *Beaulieu, 2002*; *Hubas et al., 2006*) while the contribution of larger fauna is often assumed to be relatively small (*Rowe et al., 2008*).

Benthic macro- and megafauna contribute to sediment community oxygen consumption (SCOC) both directly through respiration and indirectly through processes such as feeding, defecation, enzyme release, and bioturbation that stimulate bacterial metabolism and promote the chemical oxidation of reduced solutes (*Riemann & Helmke, 2002*; *Lohrer, Thrush & Gibbs, 2004*; *Papaspyrou, Thessalou-Legaki & Kristensen, 2010*; *Bonaglia et al., 2014*). Studies in coastal habitats have shown that the main contribution of infauna to SCOC is mediated by these indirect effects and that infaunal respiration itself makes a relatively small contribution (*Glud et al., 2000*; *Glud et al., 2003*). Thus, the overall contribution of the infauna to deep-sea ecosystem function is most likely underestimated by allometric equations used to derive the direct contribution of fauna to overall oxygen consumption based only on body size (e.g., *Rowe et al., 2008*; *Leduc, Pilditch & Nodder, 2016*). Analyses comparing diffusive oxygen uptake, which is calculated from vertical oxygen gradients in the sediments and is thought to provide a measure of oxygen consumption by microorganisms, and total oxygen uptake based on changes in oxygen concentrations in overlying water during incubations, suggest that fauna-mediated respiration accounts for an average of about 40–60% of total benthic oxygen consumption on the upper continental slope (*Glud, 2008*). However, uncertainty remains about the interpretation of diffusive oxygen flux measurements, and the relative contributions of bacteria and other organisms to deep-sea benthic metabolism is still a matter of debate (*Rowe & Deming, 2011*). Moreover, the contribution to total oxygen uptake of large burrowing macroinfauna living deep in the sediments and in relatively low densities may be underestimated in incubations which are typically based on small areas of sediments (0.01–0.1 $m^2$).

Burrowing shrimp are common in soft sediment environments of temperate and tropical regions, and their burrowing and feeding activities mix surface and subsurface sediment resulting in substantial sediment turnover (*Stamhuis, Schreurs & Videler, 1997*; *Berkenbusch & Rowden, 1999*; *Papaspyrou, Thessalou-Legaki & Kristensen, 2004*). The presence of burrowing shrimp in coastal systems results in a 70–80% increase in sediment oxygen demand compared to sediment without shrimp, most of which is due to chemical oxidation reactions and increased microbial respiration (*Ziebis et al., 1996*; *Webb & Eyre, 2004*). Burrowing shrimp are likely to impact benthic metabolism in deeper environments such as the upper continental slope where they may also be common (*Sakai & Türkay, 1999*; *Sakai, 2005*); however, no studies have been conducted on their ecology or contribution to ecosystem function in the deep sea. Here, we estimate for the first time the direct and indirect contributions of the burrowing shrimp *Eucalastacus* cf. *torbeni* to sediment community oxygen consumption based on incubations of sediment cores from the upper continental slope of New Zealand.

## METHODS

The Chatham Rise is a submarine ridge that extends eastwards from the South Island of New Zealand at depths ∼250–3,000 m. It lies under the Subtropical Front (STF), a region where warm subtropical surface water to the north meets cold, high nutrient-low

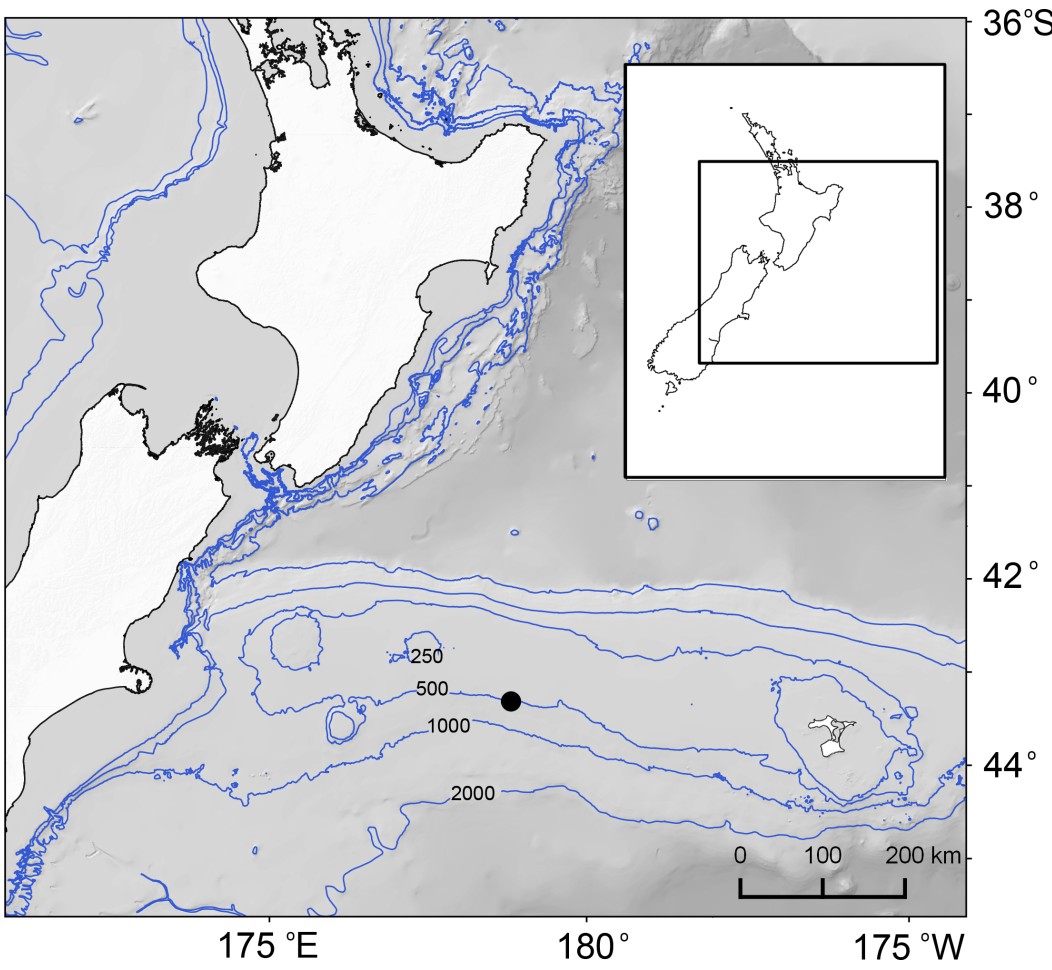

**Figure 1** Map of study area east of New Zealand's South Island and the position of sampling site (black filled circle) on Chatham Rise and 250, 500, 1,000 and 2,000 m water depth contours.

chlorophyll subantarctic surface water to the south (*Boyd et al., 1999*), which is associated with heightened primary productivity (*Bradford-Grieve et al., 1997*; *Murphy et al., 2001*). Five undisturbed sediment core samples were obtained at a site at 489 m water depth on the southern flank of the rise (43.8533°S, 178.5783°E) (Fig. 1). The samples were collected during a single deployment of an Ocean Instruments MC-800A multicorer (MUC; core internal diameter = 9.52 cm) in August 2015 (NIWA voyage TAN1511, station 181). The samples were collected under Special Permit (542) issued by the Ministry for Primary Industries pursuant to section 97(1) of the Fisheries Act 1996.

Estimates of sediment community oxygen consumption (SCOC; in $\mu$mol $O_2$ m$^{-2}$ h$^{-1}$) were obtained using shipboard incubations. Details of the incubation set-ups and measurement protocols are provided in *Nodder et al. (2007)* and *Pilditch et al. (2015)*. Briefly, the upper 13–15 cm of sediment and the overlying water from undisturbed multicore tubes were carefully extruded into transparent plastic incubation chambers (total volume = 2.0 L) with the same internal diameter. Chambers were then sealed and placed in water baths at ambient bottom water temperature (7.3 $\pm$ 0.1 °C) where they were held in the dark for 28–39 h. A

magnetically driven impeller fitted to the chamber lids gently circulated water during the incubations. Approximately 6 h after the chambers were placed in the water bath, $O_2$ concentrations were measured with a pre-calibrated PreSens MICROX I micro-optode. Four to six more $O_2$ measurements were made during the incubation period, which was terminated when the initial concentrations had decreased by ~15%. SCOC was estimated from the decline in $O_2$ concentration with time (linear regression, $r^2 > 0.95$).

Sediment pigment concentrations (i.e., chlorophyll-a and phaeopigment content) were determined to provide a measure of food availability in the incubation chambers. Immediately after the incubations, the overlying water in the incubation chambers was carefully siphoned out and a sediment sample was obtained using a subcore (internal diameter = 18 mm) to a depth of 5 cm. Sediment samples were kept frozen at −80 °C and pigment concentrations were determined in duplicate using standard techniques (*Nodder et al., 2003*; *Nodder et al., 2011*).

Following subcoring for pigment analyses, the remaining sediment was processed for macro-infaunal analyses to help determine any potential differences between cores with and without shrimp. The sediments were sieved onto a 300 μm sieve at sea and fixed in 5% formaldehyde. Samples were sorted using a dissecting microscope and the abundance of major taxa (e.g., polychaetes, amphipods, ophiuroids) was quantified.

Shrimp respiration was estimated based on the allometric equation of *Mahaut, Sibuet & Shirayama (1995)* relating respiration rate ($R$, d$^{-1}$) to individual dry weight ($W$, mg C):

$$R = aW^b$$

where $a = 7.4 \times 10^{-3}$ and $b = -0.24$. Shrimp carbon weight was determined by assuming a wet:dry weight ratio of 4 and a dry:carbon weight ratio of 2.5 (*Salonen et al., 1976*). The constants $a$ and $b$ were derived by *Mahaut, Sibuet & Shirayama (1995)*, who conducted a linear regression of all published respiration rates of deep-sea organisms. Shrimp respiration ($T$), expressed as the mass (mg) of carbon dioxide ($CO_2$) released d$^{-1}$, was estimated by multiplying shrimp dry weight ($W$, mg C) by the mass-dependent respiration rate ($R$, d$^{-1}$):

$$T = W \times R.$$

Because the equation of *Mahaut, Sibuet & Shirayama (1995)* is based on measurements conducted at 2–4 °C, and the incubation was conducted at a higher temperature (7.3 °C), estimated shrimp respiration was adjusted assuming a temperature coefficient ($Q_{10}$) of 2. Shrimp respiration was converted to oxygen ($O_2$) consumption assuming that one mole of $O_2$ is consumed for each mole of $CO_2$ released (*Hargrave, 1973*).

## RESULTS AND DISCUSSION

A total of five undisturbed cores were recovered, which consisted of sandy silt with small amounts of dark glauconite particles typical of the central Chatham Rise (*Cullen, 1967*; *Orpin et al., 2008*). A small burrow with an opening approximately one centimeter in diameter was present in the center of one of the cores, which was otherwise similar in appearance to the other cores. At the end of the incubation, which lasted 28 h for the core containing

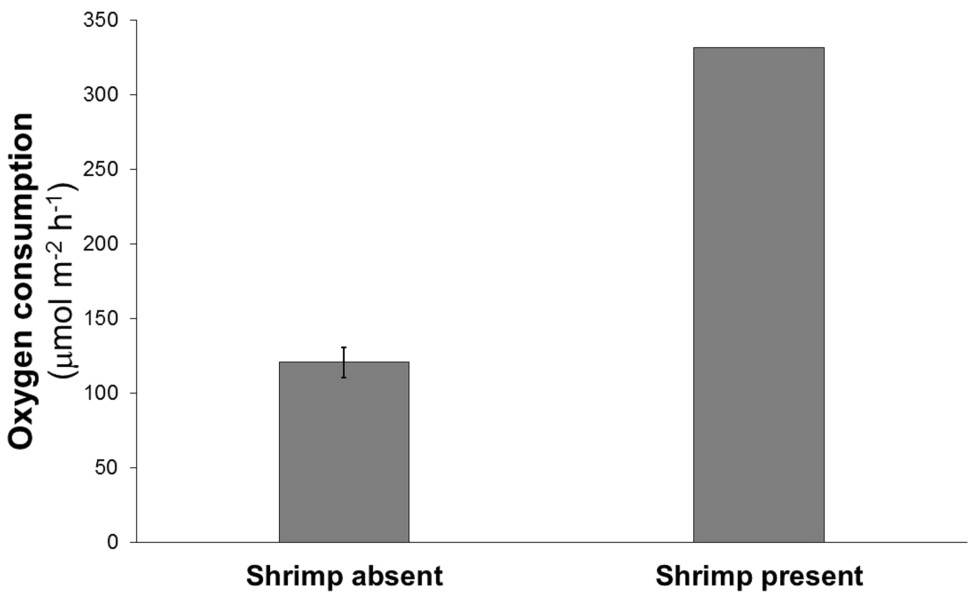

**Figure 2** **Mean oxygen consumption in four cores without shrimp and one with a single live *Eucalasta-cus* cf. *torbeni* shrimp specimen.** Error bars represent the 95% confidence interval.

the burrow opening, the presence of a live shrimp was noted for the first time. The shrimp was later identified as *Eucalastacus* cf. *torbeni*, with a length of 12 mm and wet weight of 3 mg. A gently sloping burrow of about eight millimetres in diameter, reaching to a depth of 6 cm below the sediment surface and with two branches leading to burrow entrances on the edge of the chamber, were visible through the transparent chamber wall. The original burrow opening in the center of the core was still present and was connected to the two new openings by the recently built burrow. The total length of the burrow was estimated as 27 cm, corresponding to a surface area of about 68 cm$^2$. The surface area of an undisturbed (flat) sediment surface in the incubation chamber is 71 cm$^2$, therefore burrow formation by the shrimp led to a doubling of the surface area of the sediment-water interface. The shrimp was intact and moved freely in the burrow; brown particles could be seen in its intestine, suggesting that it had recently been feeding.

Oxygen consumption in the chamber containing the shrimp was about three times greater than the mean oxygen consumption of the chambers without shrimp (332 vs 110–134 $\mu$mol m$^{-2}$ h$^{-1}$) (Fig. 2).

Phaeopigment concentration in the shrimp incubation chamber was similar to concentrations in the chambers without shrimp (3.3 vs 2.1–4.4 $\mu$g/g$_{sediment}$), whereas chlorophyll-a concentration was two to three times lower in the incubation chamber with shrimp than in chambers without shrimp (0.18 vs 0.47–0.62 $\mu$g/g$_{sediment}$). These findings suggest that variation in food availability among the incubation chambers is unlikely to account for the elevated oxygen uptake associated with the presence of burrowing shrimp; on the contrary, it appears that the feeding and burrowing activities of the shrimp may have led to a decrease in chlorophyll-a concentrations in the top five centimeters of sediment relative to

incubation chambers without shrimp. Similar decrease in sediment chlorophyll-a concentrations associated with the presence of burrowing shrimp have been observed in intertidal field experiments, presumably as a result of ingestion or burial (*Webb & Eyre, 2004*).

The abundance of macro-infauna in the shrimp incubation chamber was slightly higher than in the chambers without shrimp, but the difference was small (152 vs 54–151 individuals). Polychaetes were dominant in all chambers and accounted for 46–63% of total macro-infaunal abundance. The next most abundant taxa were amphipods (7–17% of total abundance) and nematodes (9–14%). Although biomass could not be determined in the present study, no obvious difference in size of macro-infaunal organisms were apparent among the cores. Thus, the elevated oxygen consumption associated with the presence of shrimp is unlikely to be due to differences in the associated macro-infaunal community.

Based on the allometric equation of *Mahaut, Sibuet & Shirayama (1995)* (and assuming a temperature coefficient of 2 to account for temperature difference), respiration by *Eucalastacus* cf. *torbeni* (0.59 $\mu$mol d$^{-1}$) accounted for about 0.8% of total respiration in the incubation chamber, and just over 1% of the average difference in oxygen uptake between incubation chambers with and without shrimp. This result is consistent with previous results suggesting that the majority of the increase in oxygen consumption associated with the presence of burrowing shrimp is due to increased oxidation reactions and/or microbial respiration resulting from the shrimp's burrowing and feeding activities (*Koike & Mukai, 1983*; *Webb & Eyre, 2004*). Our estimates suggest that the presence of one *Eucalastacus* cf. *torbeni* individual in the sediment is responsible for an oxygen uptake rate of about 1.7 $\mu$mol h$^{-1}$ or 40 $\mu$mol d$^{-1}$. This value is about 20 times less than the oxygen uptake resulting from the presence of a single burrowing *Callianassa japonica* or *Trypea australiensis* individual in subtidal environments (*Koike & Mukai, 1983*; *Webb & Eyre, 2004*) which could reflect the larger body size and burrows of the latter species relative to *E.* cf. *torbeni* and/or the comparatively low metabolic rates of organisms living in cold deep-sea environments. The relatively deep oxygenated layer of deep-sea sediments combined with low organic matter concentrations may also limit the extent to which bioturbation can stimulate oxygen uptake compared to shallow environments.

Data on sediment community oxygen consumption, which are typically based on *in situ* or onboard incubations of small sediment cores that do not include large burrowing fauna, are likely to be underestimating true benthic metabolism rates in deep-sea habitats (*Glud, 2008*). No estimates are available on the population densities of burrowing shrimp in deep-sea environments; however in shallow environments burrowing shrimp densities range from a few individuals to several hundred individuals per square meter (*Berkenbusch & Rowden, 1998*; *Dumbauld, Armstrong & Feldman, 1996*). Images of soft sediment habitats on Chatham Rise often show high densities of burrows and mounds, which are consistent with a high abundance of burrowing shrimp and other bioturbating macro-infauna (Fig. 3). However despite the apparently wide distribution of *Eucalastacus* cf. *torbeni* on New Zealand's continental margin (S Mills, pers. comm., 2016), no data are available on their densities due to the paucity of sufficiently large and/or quantitative sediment samples. Based on our findings, the presence of ten burrowing shrimp per square meter of seabed, a

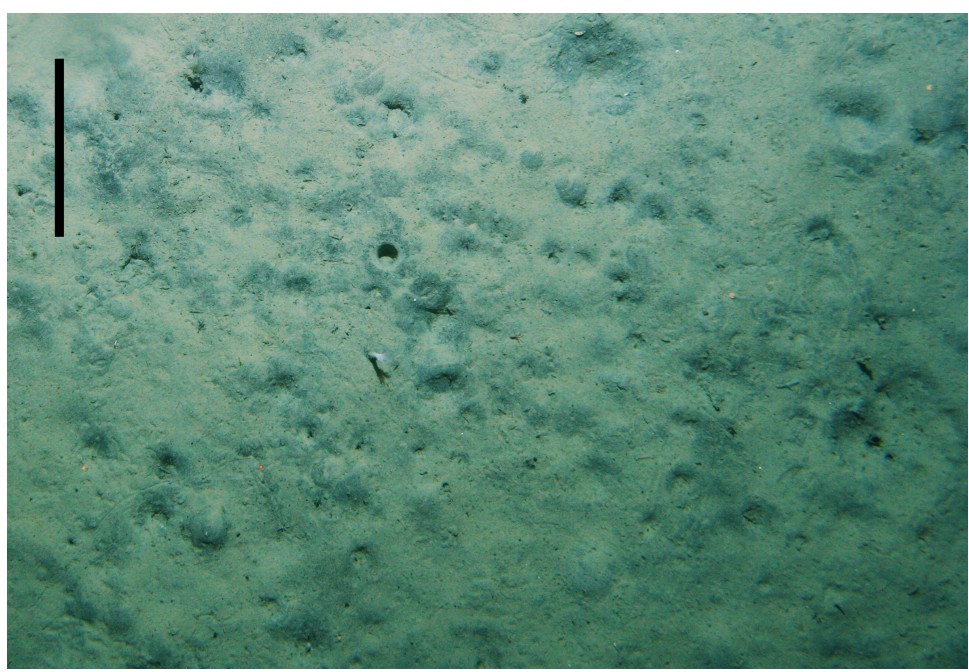

**Figure 3** **Picture of the seabed taken using NIWA's Deep Towed Imaging System (DTIS) on 13 June 2013 (*RV Tangaroa* voyage TAN1306, station 69) taken approximately 300 m away from study site.** Small burrows and mounds can be seen to occur at high densities. The surface area of the image is approximately 1.25 m$^2$ and the scale bar = 20 cm.

moderate shrimp density broadly consistent with burrow densities in seabed images in the region, would increase oxygen uptake by 17 μmol m$^{-2}$ h$^{-1}$, equivalent to 14% of current estimates obtained from sediment core incubations from the study site. This density of burrowing shrimp would translate to an oxygen uptake comparable to current estimates of total macro-infaunal community respiration on Chatham Rise based on allometric equations (2–23 μmol m$^{-2}$ h$^{-1}$ *Leduc, Pilditch & Nodder, 2016*).

Besides shrimp, other burrowing macro- and megafauna, such as echiurans, holothurians and ophiuroids, are likely to make a substantial indirect contribution to sediment community respiration. Because these relatively large organisms are vulnerable to physical disturbance from human activities such as bottom trawling (*Clark et al., 2016*), a decrease in their density and/or diversity would likely result in a loss in ecosystem function. Evaluating the magnitude of this loss, which is likely to be greatest in relatively high productivity upper continental slope habitats where large benthic fauna are most abundant (*Leduc, Pilditch & Nodder, 2016*), may be aided by the application of functional trait assessment approach (*Bremner, Rogers & Frid, 2003*) and the use of *in situ* O$_2$ consumption measurement method capable of integrating larger and therefore more representative areas of the seabed, such as *in situ* benthic chamber incubations (*Lichtschlag et al., 2015*) or the eddy correlation flux method (*Berg et al., 2009*).

## ACKNOWLEDGEMENTS

We thank the scientific personnel, officers and crew of RV *Tangaroa* of NIWA voyage TAN1506. We are also grateful to Nathania Brooke (University of Waikato) for her help with the incubations, Jeff Forman (NIWA) for shrimp identification, Sadie Mills (NIWA) for providing *Eucalastacus* cf. *torbeni* distribution data, and Ashley Rowden and Matt Pinkerton (NIWA) for their advice and support. We thank Clifton Nunnally and an anonymous reviewer for their constructive criticisms on the manuscript.

### Funding

This research was funded by NIWA under Coasts and Oceans research programme Ecosystem Structure and Function. The funders had no role in study design, data collection and analysis, decision to publish, or preparation of the manuscript.

### Grant Disclosures

The following grant information was disclosed by the authors:
NIWA under Coasts and Oceans research programme Ecosystem Structure and Function.

### Competing Interests

The authors declare there are no competing interests.

### Author Contributions

- Daniel Leduc conceived and designed the experiments, performed the experiments, analyzed the data, wrote the paper, prepared figures and/or tables.
- Conrad A. Pilditch contributed reagents/materials/analysis tools, wrote the paper, reviewed drafts of the paper.

### Field Study Permissions

The following information was supplied relating to field study approvals (i.e., approving body and any reference numbers):

The samples were collected under Special Permit (542) issued by the Ministry for Primary Industries pursuant to section 97(1) of the Fisheries Act 1996.

### Data Availability

The raw data has been supplied as Data S1.

### Supplemental Information

Supplemental information for this article can be found online at http://dx.doi.org/10.7717/peerj.3309#supplemental-information.

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
