# Peer review of "Estimating the effect of burrowing shrimp on deep-sea sediment community oxygen consumption"

_PeerJ, doi:10.7717/peerj.3309_

## Round 0.1 · original submission · Minor Revisions

The paper is well reviewed and minor changes only are needed. Both reviewers mention Q10.

·

Basic reporting

• The authors used the collected cores of a single multi-core drop to conduct an impromptu investigation of megafauna burrow (specifically a shrimp) presence on the total consumption of oxygen by a sediment community. Six cores were collected with one containing a species of burrowing shrimp and thus provided for a 5 to 1 comparison of total sediment respiration, with and without a large burrowing crustacean.
• The use of acceptable equations for determining individual respiration from Mahuat et al. 1995 allowed the authors to distinguish the results of individual oxygen consumption from the impact of bioturbation and bioirrigation. This is an important distinction important for separating potential confounding factors.

Experimental design

• Although the experimental techniques used were valid and conducted expertly, the study is not particularly robust as there is no replication and only one core contained a shrimp. This however is common in deep-sea biology because the investigator is often not able to choose how cores will look or what they may sample.
• The depth of the study area is shallow enough so that ex situ incubation of the cores does not introduce serious bias of results.
• Included in the data set should have been the calculated volume of the overlying water in each of the core incubations in addition to the height of overlying water.
• To better constrain the results a Q10 transformation of the respiration rates for both the sediment and shrimp are expected. I strongly recommend that this be done in the revision of the manuscript, as it is important for use for future research lines.

Validity of the findings

• Although this study is small and the number of experiments do not lead to a robust data set the information is extremely important for reasons outline in the introduction (Lines 55-56).
• The validity of the results is consistent with the known literature. Extreme care and thought given to the experiment and the intricate analysis of the data make the findings well founded.

Additional comments

• I find that this is an excellent manuscript and a good example of adaptability in research questions. It is true that it is short but the data presented are important and the results are a key step forward in determining the link between habitat variability and ecosystem function in the deep-sea. I applaud the authors’ rigorous use of previous work and their excellent experimental rigor. I hope that a more detailed investigation along these lines are now in planning for the future of research on the Chatham Rise.

Reviewer 2 ·

Basic reporting

.

Experimental design

.

Validity of the findings

.

Additional comments

Review of "Estimating the effect of burrowing shrimp on deep-sea sediment community oxygen consumption" by D. Leduc and C. Pilditch for PeerJ.

This is a nice, short ms. that is to the point, makes a point and then ends: all of which I like. These findings are useful but not earthshaking. This is something that would have been expected.

A nice addition would have been more information on the habitat; it is probably in the Leduc et al. ms. cited. A brief review of the depth gradients of biomass, densities, etc., would have added to the ms, but now the reader can go to the cited paper for that information.

I have one quarrel with the general discussion that is in the abstract. In line 20 they say the consensus is that bacteria drive the processes: I think that is an open question. The data are generally lacking. For example, see the paper by Rowe, G. and J.Deming. 2011. [An alternative view of the role of heterotrophic microbes in the cycling of organic matter in the deep sea. Marine Biology Research 7(7), 629-636. doi: 10.1080/17451000.2011.560269] They compare measured bacterial rates with total sediment oxygen uptake and conclude that the bacteria do not dominate because they need the macrofauna and meiofauna to produce DOC from POC. This idea bears consideration. This so-called consensus is repeated in their reference to two of Glud's papers, but, again, the data are not there. True, the biomass of the microbiota is very high, but most are probably inactive. Likewise, they reference Rowe et al., 2008, but that must have been written prior to the 2011 paper with Deming: they must have changed their minds after looking at all of their data. To me, the issue is still open.

In lines 201 to 208 they compare their estimates with the allometric estimates made by Mahaut. But the temperatures were different. This needs clarification. They could impose a Q10 effect, either lowering or raising their rates by about 50%. A Q10 of 2 would mean that the rate doubles every 10 degree rise in temperature. With a 5 degree difference, this would be about 50%. So, they could consider raising or lowering either their value or Mahaut's value. This does not alter their basic conclusions: holes in the sediment increase the total surface area and thus the oxygen exchange is increased. The more holes and the more pumping by animals in the holes, the greater the exchange of oxygen across the interface. This is THEIR IMPORTANT POINT.

Two minor issues: line 87 and line 228: 'however' is not a coordinating conjunction and thus they need a semi-colon instead of a comma at these junctures.

This ms. deserves to be published with minor revisions.

---

## Round 0.2 · accepted · Accept

Thank you for your revision and attention to the reviewers' comments.